# Spatiotemporal Predictive Pre-training for Robotic Motor Control

## Abstract

Robotic motor control necessitates the ability to predict the dynamics of environments and interaction objects. However, advanced self-supervised pre-trained visual representations (PVRs) in robotic motor control, leveraging large-scale egocentric videos, often focus solely on learning the static content features of sampled image frames. This neglects the crucial temporal motion clues in human video data, which implicitly contain key knowledge about sequential interacting and manipulating with the environments and objects. In this paper, we present a simple yet effective robotic motor control visual pre-training framework that jointly performs spatiotemporal prediction with dual decoders, utilizing large-scale video data, termed as **STP**. STP adheres to two key designs in a multi-task learning manner. First, we perform spatial prediction on the masked current frame for learning content features. Second, we utilize the future frame with an extremely high masking ratio as a condition, based on the masked current frame, to conduct temporal prediction of future frame for capturing motion features. This asymmetric masking and decoder architecture design is very efficient, ensuring that our representation focusing on motion information while capturing spatial details. We carry out the largest-scale BC evaluation of PVRs for robotic motor control to date, which encompasses 21 tasks within a real-world Franka robot arm and 5 simulated environments. Extensive experiments demonstrate the effectiveness of STP as well as unleash its generality and data efficiency by further post-pre-training and hybrid pre-training. Our code and weights will be released for further applications.

## 1 Introduction

In NLP and CV, adapting pre-trained foundation models from large-scale data to various downstream tasks has seen great success. For example, pre-trained visual representations using self-supervised [38, 15, 67, 2, 93] or weakly-supervised [71, 25, 55] methods exhibit strong generalization ability for visual understanding. However, in robot learning, due to data scarcity and homogeneity, some groundbreaking methods [53, 1] resort to training from scratch only using domain-specific data. Recently, inspired by the success of transfer learning in CV, many works [69, 73, 65, 58, 59, 19] have explored developing a pre-trained visual representation (PVR) using large-scale out-of-domain data for various robotic motor control tasks. Currently, one successful paradigm [73, 99, 59, 19] is to use large-scale egocentric video datasets [29] and train vanilla vision transformers (ViT) [22] based on MAE [38], which exhibits excellent learning efficiency and generalization ability for learning policy from raw pixel. Among them, the Ego4D [29] dataset offers numerous first-person human-object interaction scenes and good motion clues. We argue that although learning static spatial structure priors from task-relevant pre-training data sources is crucial, designing a more relevant self-supervised proxy task for motor control should not be overlooked. Therefore, in this paper, we aim to develop a more relevant self-supervised proxy task for robotic motor control representation learning.

Submitted to 38th Conference on Neural Information Processing Systems (NeurIPS 2024). Do not distribute.

Robotic motor control typically requires fine-grained spatial localization and relatively dense semantics. With its ability to effectively capture low-level geometry and space structure, MAE [38] pre-training excels at this task. However, is dense spatial content sufficient for robotic motor control? Some neuroscientific studies [50, 21, 88] suggest the brain's different areas or cells show specialization. Some are dedicated to processing the information of temporal object motion, while others focus on static spatial details. Their combination results in subjective pattern perception. Inspired by this finding, we hypothesize that an effective robotic motor control pre-training proxy task should require joint learning of spatial content features and temporal motion features. However, current methods [73, 59, 19] use MAE pre-training with image frames from human videos, capturing only static content features. They overlook the temporal motion clues in human videos, which implicitly contain key knowledge about sequential interaction with environment and manipulation of objects. Therefore, we aim to bridge this gap by incorporating these motion clues into our proxy task.

Based on the analysis above, the most critical challenge is the absence of action annotations in human video data for modeling object motion. To model interaction and manipulation actions from actionless video data, we propose to implicitly capture them by predicting future frame pixels based on current frame. However, predicting the future frame without any conditions could contain high uncertainty and be extremely difficult. Therefore, we propose to use the future frame with an extremely high masking ratio as a prompt condition, specifically 95%, which serves to reveal some behavior and dynamic priors, i.e. what to do and how to do it. In the experiments section, we will further explore different condition alternatives, including language narration and their combination. Additionally, directly and simply executing temporal prediction could lead the model to overlook static spatial details, and it is also not efficient enough. Therefore, another technical contribution of STP is to jointly perform spatial prediction by masking the current frame with 75% masking ratio. In summary, we present **STP**, a multi-task self-supervised pre-training framework through spatiotemporal predictive learning. Our STP asymmetrically mask the current frame and future frame from a video clip, using a spatial decoder to conduct spatial prediction for content learning and a temporal decoder to conduct temporal prediction for motion learning. This asymmetric masking and decoder architecture design ensures that our pre-trained encoder focusing on motion information while capturing spatial details.

Subsequently, we establish our evaluation scheme. Currently, how to adapt pre-trained visual representations for robotic motor control still remain an open question. Considering the expensive cost of robot data collection or exploration, we employ a data-efficient paradigm of few-shot behavior cloning by learning from demonstrations (Lfd). To demonstrate the generalization ability of visual representation, our primary evaluation scheme involves freezing the visual encoder during policy training. Additionally, considering that fine-tuning ViT with few demonstrations might lead to overfitting and masked modeling exhibits excellent data efficiency [86, 102, 52] in domain-in data, we further follow the post-pre-training [7, 93, 59] paradigm to perform STP pre-training with task-specific data to achieve better results. It is noteworthy that different tasks do not share representation in this setting. Finally, we conduct the largest-scale BC evaluation of PVRs for robotic motor control to date to demonstrate the effectiveness of STP, which encompasses 21 tasks ( 2 real-world tasks and 19 simulation tasks across 5 environments). These simulation tasks are derived from the union of manipulation and locomotion tasks from prior works [65, 59].

We make the following **four contributions**: **(1)** We present STP, a *self-supervised* visual pre-training framework for robotic motor control, which jointly conducts spatiotemporal prediction with *asymmetric masking and decoder architecture design* for content and motion features learning. **(2)** We further expand STP by performing hybrid pre-training with ImageNet-MAE and post-pre-training with task-specific data, unleashing its *generality* and *data efficiency*. **(3)** To our best knowledge, we conduct the *largest-scale BC evaluation* of PVRs for robotic motor control to date to demonstrate the effectiveness of STP. **(4)** Our experiments yield some insightful observations. In temporal prediction, language does not significantly enhance performance. Instead, *single-modality self-supervised paradigm* achieves the best results. This finding is highly encouraging for self-supervised robotic motor control representation learning. Moreover, in the few-shot BC setting, naively scaling up model size does not necessarily lead to improved outcomes. Finally, incorporating *more diverse* data and *domain-in* data into the pre-training can further enhance performance.

## 2   Related Work

**Pre-trained Visual Representation Learning.** Large-scale visual representation pre-training are continually empowering computer vision. The primary supervised learning methods include learning

image recognition [40, 87] from ImageNet [20] and learning multi-modal alignment [71] from image-text pairs. Currently, self-supervised learning methods are enjoying significant popularity, primarily falling into two main categories. The first category utilizes contrastive learning [39, 15, 14] technique or joint-embedding architecture [13] to learn view-invariance. The second category performs masked modeling [7, 38, 100, 95, 4, 2] and predict the pixel or representation of invisible parts in space. In addition, some methods [106, 67, 8] have also proposed to combine different optimization objectives in a multi-task learning manner. Recently pre-trained visual representation learning for robotic motor control have bee rapidly developing [69, 65, 73, 99, 58, 57, 46, 59, 19]. These methods cover different backbones (ResNet [40], ViT [22]), different policy learning methods (reinforcement learning [99], behavior cloning [69, 65, 59], reward function [58] and task specification [42]), different adaptation schemes (linear probing [69, 65, 46, 59], fine-tuning [19] and designing adapters [78, 56]), and different evaluation environments (diverse simulation benchmarks). At present, it is still unclear how these factors collectively influence the performance. In this paper, we choose scalable vanilla vision transformer [22] as our backbone and data-efficient few-shot behavior cloning paradigm to conduct policy learning, while ensuring the backbone is frozen during policy training.

**Temporal Predictive Learning.** Early works once explored representation learning through future prediction, encompassing image [61], video [35, 80] and audio [66]. VideoMAE [86, 93] extend MAE [38] to 3D video architecture. Recently TrackMAE [17] and SiamMAE [33] predict the masked future frame based on unmasked current frame, leading to a better capture of temporal correspondence and achieving outstanding performance in object tracking and segmentation tasks. In robot learning, predicting future visual states primarily serves as a transition dynamic model such as World Models [62, 77] and Dreamer [76]. [85, 9] predict the future visual states using goal image in robot data. GR-1 [97] conducts language-conditioned video prediction for policy model pre-training in a frozen visual representation space. [96] proposed dynamics-aware representation learning, and [82, 72] employed forward dynamics for self-supervised pre-training. Some works explored to train video prediction models and utilize visual foresight [32], inverse dynamics models [18], goal-conditioned policy learning [23], and geometry estimation [51] methods for motor control, respectively. [92] fine-tuned pre-trained representations into dynamic and functional distance modules for manipulation tasks. Unlike these works, we utilize the public large-scale egocentric video data and employ masked spatiotemporal predictive learning as a *self-supervised proxy task (without any language or action annotations) for robotic motor control representation learning*, instead of designing elaborate architectures or methods for specific predictive tasks [28, 37].

**Vision-based Robot Learning.** Vision-based robot learning plays a crucial role in robotics community. Recently some related works focus on studying model architectures [44, 12, 47], observation spaces [107], downstream policy learning methods [41], sim-to-real transfer [79], designing adapters [78, 56], learning-from-scratch baseline [36], and affordance model [6, 105, 45, 60], in visuo-motor representation learning. Other related works [70, 5, 91, 101, 48] attempt to learn manipulation skills from small-scale and in-domain human videos. In addition, language-conditioned vision robot learning has received significant attention. Some works scale multimodal robotic data [42, 11, 34, 90, 24, 68, 84] or introduce Internet data and knowledge [81, 103, 10, 54, 43, 94, 64] for end-to-end robot learning. In our study, we pre-train a off-the-shelf visual representation from large-scale egocentric video datasets for robotic motor control tasks. Our method is more simple and general for different downstream tasks of motor control.

# 3 Method

In this section, we describe our method in details. First, we give an overview of our spatiotemporal predictive pretraining (STP) framework. Then, we give a technical description on our core components during pre-training: the masked image encoder and dual decoders scheme. Finally, we describe how to adapt our pre-trained encoder to downstream robotic motor control tasks.

## 3.1 Overiew of STP

As illustrated in Figure 1, our STP aims to pre-train an image encoder for robotic motor control from video datasets. This pre-trained image encoder is subsequently frozen and directly transferred to solve motor control tasks. Specifically, given a video dataset $\mathcal{D}$, our goal is to learn an image encoder $\Phi_{enc}$, that maps images to the visual representations. During pre-training and post-pre-training, $\mathcal{D}$ represents large-scale out-of-domain videos and task-specific demonstration videos, respectively. After pre-training, we reuse $\Phi_{enc}$ for downstream motor control policy learning. Specifically, the downstream

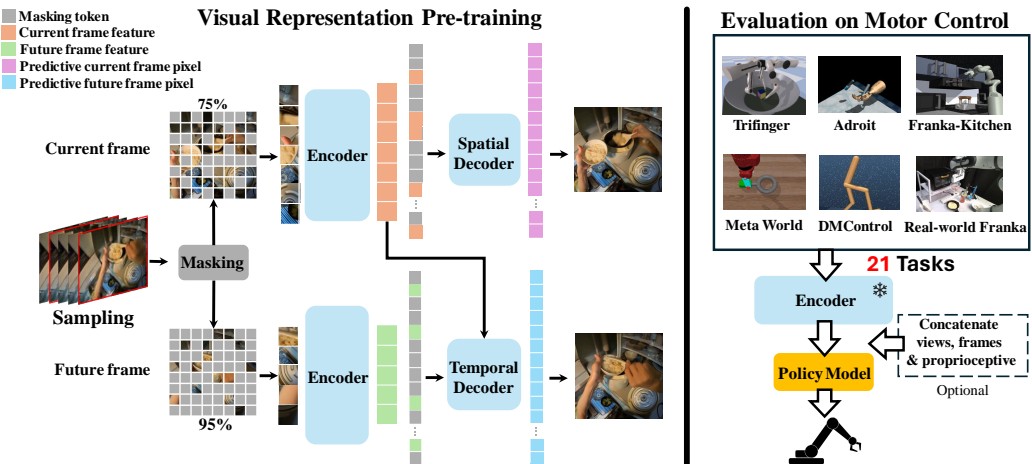

Figure 1: **STP framework**. **Left:** During pre-training, we sample the current frame and the future frame from the video clip, and carry out spatiotemporal predictive pre-training. **Right:** During motor control tasks evaluation, we freeze the pre-trained encoder to extract visual state representations and discard the decoders.

task will require an agent to make sequential action decisions based on visual observations $\mathcal{O}$. Instead of using the raw observation images as direct input like end-to-end policy learning from pixel, the agent will employ the pre-trained $\Phi_{enc}$ to extract its visual state representation $\Phi_{enc}(\mathcal{O})$ for the subsequent policy learning module.

### 3.2 Masked Image Encoder

We first introduce the pipeline of our image encoder. Our image encoder processes image frames using a vanilla vision transformer [22]. Given a image $\mathbf{I} \in \mathbb{R}^{C \times H \times W}$, we initially process it by the patch embedding layer to obtain its token sequences $\mathbf{T}$, where $\mathbf{T} = \{P_i\}_{i=1}^{N}$ and $N$ is the the total token number, (e.g., N = 196 for a 224 × 224 image with a patch size of 16 × 16). Then we add the fixed 2D sine-cosine positional embeddings for all tokens. Following this, we mask and remove a part of tokens, according to a randomly generated masking map $\mathbb{M}(\rho)$, where $\rho$ is the masking ratio. The encoder applies several transformer blocks (consisting of a global self-attention layer and a FFN layer) on all unmasked tokens: $\mathbf{Z} = \Phi_{enc}(\mathbf{T}^u)$, where $\mathbf{T}^u = \{T_i\}_{i \in (1-\mathbb{M}(\rho))}$. During this process, a [CLS] token is added at the beginning.

Then we describe our encoding process during pre-training. We randomly sample two frames from a video clip based on an interval: the current frame $\mathbf{I_c}$ and the future frame $\mathbf{I}_f$. Following the above pipeline, we randomly generate two asymmetric masking maps for the current frame and the future frame, denoted as $\mathbb{M}_c = \mathcal{M}_c(\rho^c)$ and $\mathbb{M}_f = \mathcal{M}_f(\rho^f)$, respectively. Each of these maps has a different masking ratio. We then use these maps to separately process the two frames and obtain their features, $\mathbf{Z}_c$ and $\mathbf{Z}_f$. As analyzed above, our STP aims to jointly learn content and motion features by spatiotemporal predictive learning. For content feature learning, we follow MAE [38], masking a portion of the current frame based on $\mathbb{M}_c$, with $\rho^c = 75\%$, and predict the masked parts during the decoding process. This encourages the model to learn spatial and geometric structure priors from the current frame data through spatial reasoning. For motion feature learning, we establish an objective to predict the future frame based on the masked current frame. However, predicting the future frame without any conditions could be meaningless and extremely challenging. Therefore, we use the future frame with an extremely high masking ratio as a condition, specifically $\rho^f = 95\%$, which reveals some behavior and dynamic priors. In the experiments section, we will further discuss different condition schemes, including language narration and the combination between them. In summary, our encoding process during pre-training can be formally described as follows:

$$\begin{cases} \mathbf{Z}_c = \Phi_{enc}(\mathbf{I}_c, \mathbb{M}_c), \\ \mathbf{Z}_f = \Phi_{enc}(\mathbf{I}_f, \mathbb{M}_f). \end{cases} \tag{1}$$

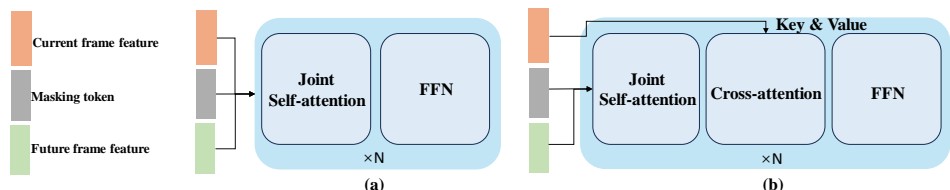

Figure 2: Temporal decoder design. **(a)** Standard joint-self architecture. **(b)** Our self-cross architecture.

## 3.3 Dual Decoders

To jointly capture static content and object motion features for better spatiotemporal understanding, our STP present a dual decoders scheme to predict both the pixel of current and future frame simultaneously in a multi-task learning manner. As shown in Figure 1, our dual decoder scheme includes a spatial decoder $\Phi_{dec\_s}$ for spatial prediction and a temporal decoder $\Phi_{dec\_t}$ for temporal prediction. We firstly give a technical description on them, respectively. Then we describe how we combine them into our final method.

**Spatial Decoder.** To capture static content features, our spatial decoder is solely utilized for processing the current frame visual feature. Specifically, after obtaining the masked current frame visual feature $\mathbf{Z}_c$, we concatenate it with some learnable masking tokens, leading to the formation of $\mathbf{Z}_c^d = \mathbf{Z}_c \cup \{\mathbf{M}_i\}_{i \in \mathbb{M}_c}$, where $\mathbb{M}_c$ is the current frame masking map. Then, each of these tokens further adds a corresponding positional embedding. Subsequently, $\mathbf{Z}_c^d$ undergoes decoding in the decoder and is continuously updated. The architecture of the spatial decoder block aligns with the standard transformer encoder block, comprised of a global self-attention layer and a FFN layer. Finally, with the deocoded token sequence $\mathbf{Z}_c^d$, our spatial decoder predicts the invisible tokens of the current frame $\hat{\mathbf{I}}_c^d$, operating under the current frame masking map $\mathbb{M}_c$.

**Temporal Decoder.** To capture motion features, our temporal decoder jointly processes the current frame and the future frame which serves as the temporal prediction condition. To elaborate, we firstly obtain the masked current frame feature $\mathbf{Z}_c$ and the masked future frame feature $\mathbf{Z}_f$. We then concatenate $\mathbf{Z}_f$ with the masking tokens that have the positional embedding added, resulting in $\mathbf{Z}_f^d$. Following this, $\mathbf{Z}_f^d$ and $\mathbf{Z}_c$ interact within the temporal decoder for decoding. The architecture of our temporal decoder block is in alignment with the standard transformer decoder block [89], consisting of a self-attention layer, a cross-attention layer, and a FFN layer, as shown in Figure 2 (b). During decoding, the self-attention layer and FFN are solely used to process $\mathbf{Z}_f^d$. For the cross-attention layer, $\mathbf{Z}_f^d$ is continuously updated as the query, while $\mathbf{Z}_c$, acting as the key and value, is kept constant. Compared to standard architecture, it ensures that the past frame representation space will not be updated in the temporal decoder and are specifically used for temporal correlation and prediction. This asymmetric interact architecture not only achieves more efficient training but also produces better results. Finally, with the decoded token sequence $\mathbf{Z}_f^d$, our temporal decoder predicts the invisible tokens of the future frame $\hat{\mathbf{I}}_f^d$, operating under the future frame masking map $\mathbb{M}_f$.

**Multi-task Predictive Learning.** As mentioned above, our STP jointly conducts spatiotemporal prediction by asymmetric masking ratio and dual decoders scheme, the whole decoding pipeline can be formally described as follows:

$$
\begin{cases}
\hat{\mathbf{I}}_c^d = \Phi_{dec\_s}(\mathbf{Z}_c^d), \\
\hat{\mathbf{I}}_f^d = \Phi_{dec\_t}(\mathbf{Z}_c, \mathbf{Z}_f^d).
\end{cases}
\tag{2}
$$

Our loss function is the mean squared error (MSE) loss between the normalized masked pixels and the predicted pixels. So our loss function $\ell$ is as follows:

$$
\ell = \mathrm{MSE}(\hat{\mathbf{I}}_c, \mathbf{I}_c) + \mathrm{MSE}(\hat{\mathbf{I}}_f, \mathbf{I}_f).
\tag{3}
$$

## 3.4 Downstream Policy Learning

To enable data and computation efficiency during the policy learning process, we adopt the paradigm of few-shot behavior cloning by learning from demonstrations (Lfd), and we keep the image encoder

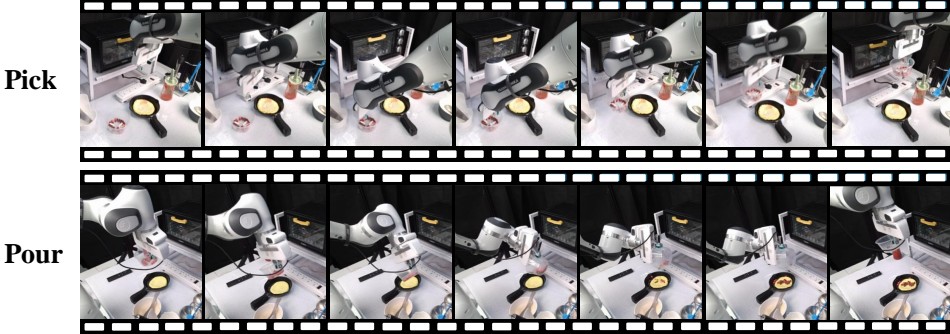

Figure 3: **The evaluation demonstrations of our real-world tasks.** For picking, the robot arm needs to pick up the bowl on the desktop. For pouring, the robot arm needs to pour the ingredients from the bowl into the pot.

frozen. Concretely, for each task, we are given offline expert demonstrations $\mathcal{S} = \{\tau_1, ..., \tau_n\}$, where each $\tau_i$ is a trajectory of robot observations and actions, denoted as $\tau_i = [(o_0, a_0), \ldots, (o_T, a_T)]$. Based on the $\mathcal{S}$, we train a policy mdoel, $\pi_\theta(a|\mathcal{C}(\Phi_{enc}(o)))$, parameterized by $\theta$, which maps from robot's state representations to actions. Here, $\mathcal{C}$ represents an optional concatenation operation that effectively fuses multi-view and multi-frame visual features, along with the robot's proprioceptive state in the channel dimension. We optimize the $\pi_\theta$ through a standard behavior cloning MSE loss:

$$\min_\theta \sum\nolimits_{(o,a)\sim\mathcal{S}} \mathrm{MSE}(a, \pi_\theta(\mathcal{C}(\Phi_{enc}(o)))). \tag{4}$$

# 4 Experiments

## 4.1 Implementation on Pre-training

We execute pre-training with data from EgoVLP [55] for comprehensive ablation and fair comparison. It processes untrimmed videos of Ego4D and filters out that miss language narrations and belong to validation or test sets, resulting in a total of 3.8 million clips, called as Egoclip. In pre-training, we sample a frame pair from each clip for training. As for all experiments, we employ ViT [22] as backbone. Additionally, we maintain consistency with prior works [73, 59], directly using the [CLS] token as the global representation. The pre-training hyperparameters can be found in section A.3.

## 4.2 Implementation on Downstream Policy

**Evaluation Scheme.** Following popular settings on PVRs for robotic motor control [65, 46, 59], for each task, we learn a single policy $\pi$ which is structured as a MLPs network. The policy models utilize both the history of visual observation embeddings and optional robot proprioceptive as inputs, subsequently generating executable actions as outputs.

**Simulation Tasks.** We select the union of manipulation and locomotion tasks from prior works [65, 59] for evaluation, encompassing 19 tasks across 5 simulated environments. These inclue Meta-World [104] (Assembly, Bin-Picking, Button-Press, Drawer-Open, and Hammer), Franka-Kitchen [31] (Sliding Door, Turning Light On, Opening Door, Turning Knob, and Opening Microwave), Adroit [74] (Relocate and Reorient-Pen), DMControl [83] (Finger-Spin, Reacher-Hard, Cheetah-Run, Walker-Stand, and Walker-Walk), and Trifinger [98] (Reach-Cube and Push-Cube). More detailed simulation evaluation details can be found in section A.4.

**Real-World Tasks.** In our real-world experiments, we evaluate contact-rich picking and pouring tasks using a Franka Emika Research 3 robot arm in a tabletop environment, ensuring no duplication with simulation Franka-Kitchen [31]. For each task, we collect 100 noise demonstrations for training, and we conduct 20 trials per task during evaluation phase. The robotic arm and objects have different initial pose between training and testing. The evaluation demonstrations of our real-world tasks is shown in Figure 3. Please see section A.5 for more real-world setup details.

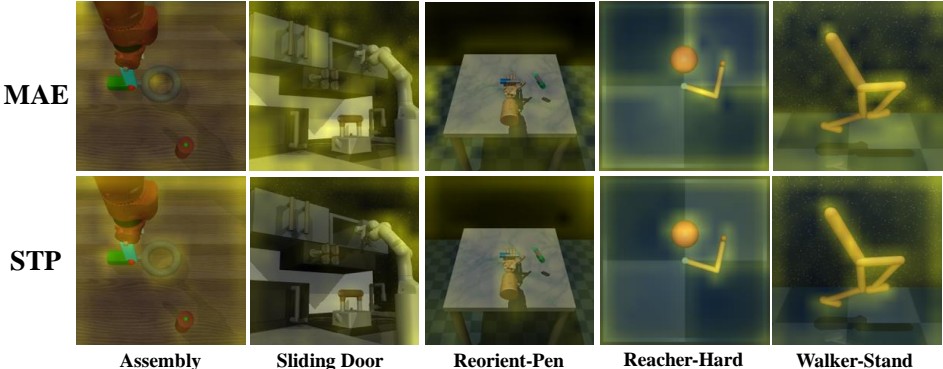

**MAE**

**STP**

Assembly  Sliding Door  Reorient-Pen  Reacher-Hard  Walker-Stand

Figure 4: **Attention Visualization.** We use the [CLS] token as query, average the attention of all heads at the last layer of the frozen ViT encoder, and perform min-max normalization. We then upsample the attention map and overlay it on the original image, where the size of the attention value is directly proportional to the intensity of the yellow light. **Top:** MAE pre-training. **Bottom:** STP pre-training.

### 4.3 Performance on Downstream Simulation Tasks

In this section, we mainly analyze the performance of some pre-trained image representations on reproducible simulation tasks. Specifically, we first evaluate the following models: (1) public DINOv2 [67] that combines masked image modeling with self-distillation on large-scale image datasets; (2) public CLIP [71] that conducts contrastive learning on large-scale image-text pairs; (3) R3M trained based on Egoclip [55]; (4) public VC-1 [59]; (5) MAE trained based on Egoclip; (6) STP trained based on Egoclip. (7) STP that conducts hybrid pre-training with initialization using ImageNet-MAE [59]. Among them, (1) and (2) achieve excellent performance on core visual understanding tasks using zero-shot or linear probing evaluation settings. (3) and (4) utilize egocentric videos for robotic motor control. (5), (6) and (7) are used for fair comparison and exploring the potential benefits of STP from more diverse image data, respectively. The experimental results are presented in Table 1. Consistent with prior findings [41, 59], there is not a universal foundation model that performs optimally across all benchmarks. However, on the whole, the MAE method is superior due to its effective modeling of low-level geometry and spatial structure, especially for the MetaWorld tasks that demand fine-grained control. Another intriguing observation is that MAE underperforms in the Franka-Kitchen and Adroit tasks. We believe that this could be due to its relatively weaker semantic representation. Under a fair comparison, our STP outperforms MAE by 4.1 (59.6 → 63.7), and additionally benefits from a more diverse image data, improving by 0.5 (63.7 → 64.2). This is attributed to that our STP not only captures static content features but also effectively models motion information by extracting temporal clues from videos of interactions and manipulations with the environment and objects. Additionally, we provide the visualization of the attention maps (model (5) and (6)) of several specific tasks in Figure 4. The results indicate that, on top of effectively capturing spatial information, our method further encourages the model to focus on motion areas or objects, thereby providing a more *sparse and compact* representation for downstream low-data BC paradigm.

Next, we also evaluate and compare the adaptation results of our representations to downstream motor control tasks. Specifically, we evaluate following settings: (a) The MAE pre-trained representation undergoes further MAE post-pre-training with task-specific data, and is frozen during policy training; (b) The STP pre-trained representation undergoes further STP post-pre-training with task-specific data, and is frozen during policy training; (c) The STP pre-trained representation undergoes end-to-end fine-tuning with task-specific data; (d) STP pre-training is performed directly using task-specific data and the resulting representation is frozen during policy training. The results show that end-to-end fine-tuning fails to yield the best results, suggesting that naively fine-tuning VIT-base could still lead to overfitting under few-shot behavior cloning scheme. Conversely, (a) and (b) achieve competitive results, with our STP achieving a 3.9 (72.5 → 76.4) improvement on the weight average success rate than MAE, further demonstrating the effectiveness and data efficiency of our STP for in-domain data. In addition, the comparison between (a) and (d) also proves the effectiveness of pre-training with out-of-domain data. Finally, we also scale up both MAE and our STP to ViT-L/16, and the results still demonstrate the superiority of STP. Among them, compared to ViT-B/16, ViT-L/16 brings a smaller performance improvement, which may be due to the task's performance saturation. However, the ViT-L/16 of STP does not show improvement in Meta-World and Trifinger, indicating that simply

Table 1: Performance comparisons of visual representations on simulation benchmarks. We report the average score across all tasks for each simulation environment. DINOv2 uses **ViT-B/14**, CLIP uses **ViT-B/32**, and unless otherwise specified, others use **ViT-B/16**. Mt-Wd, Fr-Ki, DMC, Adro, Tr-fi, and WA respectively represent MetaWorld, Franka-Kitchen, DMControl, Adroit, Trifinger, and weight average. * denotes that public VC-1 samples image frmaes form full Ego4D dataset.

| | Pre-training Data | Mt-Wd | Fr-Ki | DMC | Adro | Tr-fi | WA |
|---|---|---|---|---|---|---|---|
| DINOv2 [67] | LVD-142M | 77.9 | 41.2 | 59.4 | 50.7 | 69.0 | 59.6 |
| CLIP [71] | Image-text pairs | 75.5 | 39.8 | 52.2 | **51.3** | 57.7 | 55.6 |
| R3M [65] | Ego | 81.3 | 30.6 | 52.2 | 46.7 | 64.7 | 54.9 |
| VC-1 [59] | Ego*+MNI | 88.8 | 38.4 | 60.9 | 46.0 | 70.5 | 61.8 |
| MAE [38] | Ego | 85.1 | 36.7 | 59.2 | 43.4 | **70.6** | 59.6 |
| STP | Ego | 92.0 | 40.9 | **62.1** | 48.0 | 69.3 | 63.7 |
| STP | Ego+I | **94.1** | **42.5** | 61.6 | 47.3 | 66.7 | **64.2** |
| MAE (Post PT) | Ego+Demo | 93.6 | 46.9 | 81.1 | 58.0 | 76.8 | 72.5 |
| STP (Post PT) | Ego+Demo | **97.3** | **53.6** | **82.8** | **63.3** | **78.0** | **76.4** |
| STP (E2E FT) | Ego | 87.2 | 52.4 | 55.2 | 40.0 | 70.4 | 62.9 |
| STP | Demo | 70.3 | 30.4 | 52.5 | 38.0 | 70.8 | 51.8 |
| MAE-L/16 (Post PT) | Ego+Demo | 95.7 | 54.7 | 83.5 | 66.0 | **77.6** | 76.7 |
| STP-L/16 (Post PT) | Ego+Demo | **97.3** | **57.4** | **85.0** | **70.0** | 75.4 | **78.4** |

Table 2: The ablation experiment results. Me, Fra, DMC, Adr, Tri, and WA respectively represent MetaWorld, Franka-Kitchen, DMControl, Adroit, Trifinger, and weight average. All models use **ViT-B/16**.

(a) Current Frame Masking and Spatial Prediction.

| $\rho^c$ | Predict | Me | Fra | DMC | Adr | Tri | WA |
|---|---|---|---|---|---|---|---|
| 75% | ✓ | **92.0** | **40.9** | **62.1** | **48.0** | **69.3** | **63.7** |
| 75% | | 84.5 | 34.7 | 55.4 | 43.3 | 65.3 | 57.4 |
| 50% | ✓ | 82.1 | 36.0 | 60.3 | **48.0** | 66.8 | 59.0 |
| 0% | | 79.2 | 39.7 | 54.8 | 44.0 | 63.1 | 57.0 |

(b) Temporal Prediction Condition Design.

| Condition | Me | Fra | DMC | Adr | Tri | WA |
|---|---|---|---|---|---|---|
| L-E | 82.1 | 30.7 | 55.5 | 42.0 | 63.8 | 55.4 |
| 95% | **92.0** | 40.9 | 62.1 | **48.0** | 69.3 | **63.7** |
| 90% | 91.2 | **42.5** | 62.8 | 44.7 | 65.9 | 63.4 |
| L-E + 95% | 91.0 | 37.7 | **64.1** | 46.7 | **70.8** | 63.1 |
| L-D + 95% | 88.0 | 34.3 | 62.6 | 46.7 | 69.3 | 60.9 |

(c) Temporal Decoder Architecture Design.

| Decoder | Me | Fra | DMC | Adr | Tri | WA |
|---|---|---|---|---|---|---|
| 8 joint-self | 87.7 | 36.9 | 55.7 | 46.0 | 71.3 | 59.8 |
| 12 joint-self | 88.5 | 35.0 | 55.7 | 46.0 | 67.0 | 59.1 |
| 8 self-cross | **92.0** | **40.9** | **62.1** | **48.0** | 69.3 | **63.7** |

(d) Frame Sampling Strategy.

| Frame interval | Me | Fra | DMC | Adr | Tri | WA |
|---|---|---|---|---|---|---|
| 8 | 89.6 | 39.9 | 58.4 | 46.0 | 67.0 | 61.3 |
| 16 | 92.0 | 40.9 | **62.1** | **48.0** | **69.3** | **63.7** |
| 24 | 89.1 | **41.1** | 61.5 | 46.0 | 68.1 | 62.5 |
| 8, 24 | **92.3** | 37.1 | 57.3 | 42.0 | 68.4 | 60.8 |

scaling up model capacity does not necessarily lead to performance gains. In the few-shot BC setting, there is a risk of overfitting in both policy and backbone training.

## 4.4 Ablation on Downstream Simulation Tasks

In this section, we perform extensive ablation studies to further demonstrate the effectiveness of our joint spatial and temporal prediction, as well as temporal prediction condition design. In addition, we also study the influence of temporal decoder architecture design and future frame sampling strategy.

**Current frame masking.** The design of the current frame masking is crucial. On one hand, similar to MAE [38], masking some patches and predicting the missing parts can effectively promote the learning of image content features. On the other hand, the visible patches of the current frame need to interact with the condition to predict the future frame. Specifically, we mask the current frame at masking rates of 75%, 50%, and 0%, respectively, and optionally predict the missing parts through the spatial decoder. The results are shown in Table 2 (a). From results, we see that the masking ratio of 75% and performing spatial prediction still lead to the best performance. This demonstrates the importance of retaining MAE [38] for content features learning, especially for low-level manipulation in Meta-World, while a current frame with a high masking ratio (75%) is sufficient to interact with other conditions to predict the future frame.

**Temporal prediction condition design.** Subsequently, we discuss the influence of temporal prediction condition design. We implicitly model motion in actionless video data by predicting the pixels of the future frame. A direct and simple idea is to use language narration as a condition. The text tokens can be flexibly utilized as inputs to ViT [22], forming a multimodal encoder. Language narration

provides a high-level behavior description, but lacks low-level visual dynamic priors for pixel-level prediction. However, leaking part of the future frame can effectively provide these priors. In order to explore how to construct a more meaningful temporal prediction proxy task, we compare the following schemes: (1) only language narration, (2) masking 95% of the future frame, (3) masking 90% of the future frame, (4) masking 95% of the future frame and language narration, and (5) masking 95% of the future frame and language narration, but the language is added in the temporal decoder, instead of being fused with the visible image patches in the multimodal encoder. We tokenize all language narration by pre-trained DistilBERT [75]. The results are shown in Table 2 (b). From results, we see that using only language as a prediction condition leads to a significant decline in performance, while leaking a small amount of future frame (masking 95%) in the temporal decoder can achieve competitive results. As for joint conditions of language and future frame with 95% masking ratio, adding language in the encoder is better than in the decoder. Additionally, adding language performs better on DMControl (64.1 vs. 62.1) and Trifinger (70.8 vs. 69.3), while not adding language performs better on Meta-World (92.0 vs. 91.0), Franka-Kitchen (40.9 vs. 37.7) and Adroit (48.0 vs. 46.7). We speculate the reasons for language hurts performance are as follows: (i) The input gap (multi-modal and single-modal) between upstream and downstream; (ii) Extra language in ViT may result in the loss of some fine-grained information capture. Furthermore, the latter does not require language supervision, and can provide a more scalable self-supervised solution.

**Temporal decoder design.** We also investigate the impact of the temporal decoder design. Specifically, we consider two types of decoder blocks. One is the joint-self architecture, as shown in Figure 2 (a), and similar joint architecture are adopted in [26, 102]. The other is the self-cross architecture, as shown in Figure 2 (b), and similar cross architecture are adopted in [3, 33]. We consider the following settings: (1) 8 joint-self decoder blocks, (2) 12 joint-self decoder blocks, (3) 8 self-cross decoder blocks. Among them, setting (2) and (3) have similar amounts of parameters for a fairer comparison. The results are shown in Table 2 (c). The results demonstrate the importance of maintaining a fixed representation space of the past frame during temporal prediction.

**Frame sampling strategy.** Finally, we investigate the impact of the sampling strategy between the current frame and future frame. The difficulty of temporal prediction is directly proportional to the frame interval values. We establish four settings where we fix the sampling intervals at 8, 16, and 24 respectively, and for the fourth setting, we randomly select an interval within the range of [8, 24]. The results are shown in Table 2 (d). The results show that an interval of 16 achieves the best balance for building temporal prediction proxy task.

## 4.5 Performance on Downstream Real-world Tasks

In this section, we report our experiment results on real-world picking and pouring tasks. We report the average success rate for each task. Specifically, we compare STP with the baseline MAE, both of which are trained on out-of-domain videos and kept frozen during policy training. The results are shown in Table 3. From the results, it can be seen that STP has achieved significant advantages in the pouring task. It can more accurately align with the moving bowl

Table 3: Performance comparisons on real-world tasks.

| Method | Picking | Pouring | Average |
|--------|---------|---------|---------|
| MAE | 65.0 | 45.0 | 55.0 |
| STP | 65.0 | 65.0 | **65.0** |

and the pot. In addition, although MAE and STP have a same success rate in picking tasks, STP tends to execute grasping in a better position. This indicates that the trend and conclusion of our STP are consistent in both simulation and the real-world, which also aligns with the findings of [79].

## 5 Conclusion

In this work, we have proposed the STP, a simple, efficient and effective self-supervised visual representation pre-training framework for robotic motor control. Our STP jointly performs spatiotemporal predictive learning on large-scale videos within a multi-task learning manner. Our STP captures content features by predicting the invisible areas within the masked current frame, and simultaneously captures motion features by using a future frame with an extremely high masking ratio as a condition to predict the invisible areas within that future frame. We carry out the largest-scale BC evaluation of PVRs for robotic motor control to date to demonstrate the effectiveness of STP. Furthermore, as for pre-training data, we also prove that extending STP to hybrid pre-training and post-pre-training could further unleash its generality and data efficiency.

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

# A  Appendix

## A.1  Limitations and Discussion

Although STP has demonstrated superior performance in extensive experiments, there remain some challenges and future works. From the perspective of pre-training data, Ego4D provides numerous human-object interaction scenes and good motion clues. Building larger-scale and more diverse potential datasets such as [63, 30] to scale up STP is worth exploring. Regarding pre-training methods, exploring predictive targets outside of pixel space and more effective sampling and masking strategies present intriguing research directions. From an evaluation standpoint, we utilize a frozen ViT to extract agent state representations and adopt the paradigm of few-shot behavior cloning, other policy learning methods (reinforcement learning, visual reward function, visual task specification), have not been explored. In conclusion, as the first method of performing temporal prediction on large-scale videos for self-supervised visual representation learning intended for robotic motor control tasks, we hope STP can be taken as a strong baseline and facilitate further research along this direction.

## A.2  The influence of the loss weight ratio between temporal prediction and spatial prediction

In this section, we further explore the influence of the loss weight ratio between temporal prediction and spatial prediction. Specifically, taking five tasks from Franka-Kitchen as examples, we load the pre-trained STP and perform post-pre-training with three different loss weight ratios (temporal to spatial). The results, as shown in Figure 5, are 54.7, 55.2, and 57.4 for the average results of the ratios 3:1, 1:3, and 1:1, respectively. The results indicate that due to the different attributes of the tasks, the trends are not consistent. However, overall, the 1:1 ratio achieves the best balance and results. We chose it as a universal setting.

## A.3  Pre-training Details

In this section, we describe the details of our STP pre-training. Specifically, we list some key training and architectural hyperparameters of STP in Table 4. In addition, as for our MAE [38] baseline, we mainly follow the publicly available code of MAE[1]. Additionally, we train MAE and STP using the same data and number of epochs to ensure that the comparison between them is **completely fair**. Finally, we also provide some STP prediction results in Figure 6.

## A.4  Simulation Environments Details

In this section, we first present further details of the STP post-pre-training on downstream simulation environments. Subsequently, we delineate the specific hyperparameters used in the behavior cloning policy training within these simulation environments. Finally, we provide the comprehensive evaluation scheme for each simulation environment.

---

[1] https://github.com/facebookresearch/mae

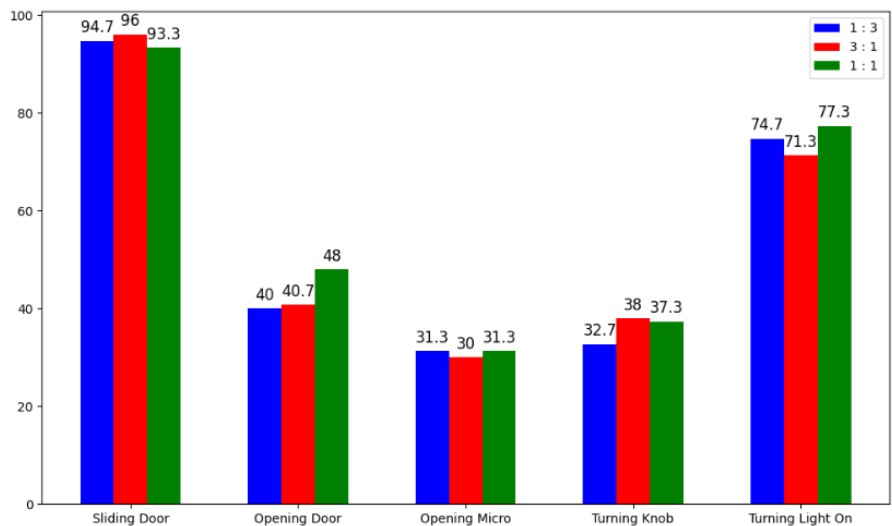

Figure 5: The results of different loss weight ratios between temporal prediction and spatial prediction.

Table 4: Training and architectural hyperparameters for STP pre-training.

| Hyperparameter | Value |
|---|---|
| *STP Pre-training* | |
| optimizer | AdamW [49] |
| base learning rate | 0.00015 |
| weight decay | 0.05 |
| optimizer momentum | $\beta_1, \beta_2 = 0.9, 0.95$ |
| effective batch size | 4096 |
| learning rate schedule | cosine decay |
| total epochs | 50 |
| warmup epochs | 5 |
| augmentation | RandomResizedCrop (0.8, 1) |
| *Encoder ViT-base Architecture* | |
| patch size | 16 |
| #layers | 12 |
| #MHSA heads | 12 |
| hidden dim | 768 |
| positional embedding | sin-cos initialization and fix |
| *Dual Decoder ViT-base Architecture* | |
| #layers | 8 |
| #MHSA heads | 16 |
| hidden dim | 512 |
| positional embedding | sin-cos initialization and fix |

In regards to the STP post-pre-training, we utilize data that aligns with the policy training, and the specific architecture hyperparameters correspond to those listed in Table 4. Depending on the specific demonstration data, we adjust the values of total epochs, warmup epochs, effective batch size, and the frame interval, as shown in Table 5.

As for policy training and evaluation schemes, we primarily refer to the publicly available code[2] and training data of VC-1 [59] for Metaworld [104], DMControl [83], Adroit [74] and Trifinger [98]. Similarly, for Franka-Kitchen [31], we follow the public code[3] and training data of R3M [65]. Specifically, the policy training hyperparameters and evaluation schemes are shown in Table 6 and Table 7, respectively. About policy training, we completely follow the setting of prior works [65, 59] when freezing the encoder; when performing end-to-end fine-tuning, we make appropriate adjustments to the batch size and learning rate. About evaluation details, similar to prior works[65, 59], we establish all evaluation details such as the number of expert demonstrations and test trajectories, environmental viewpoints, optimization hyperparameters, base seeds, history windows size, and the use of robot proprioceptive. In Table 7, the term 'prop.' stands for whether proprioceptive information is used or not, and 'history window size' signifies the number of frames received by the policy model at each step, with features between frames being fused through concatenation. 'Number of trajectories' represents the quantity of trajectories evaluated. For tasks in Meta-World, Franka-Kitchen, Adroit, and Trifinger, we report the maximum success rate, whereas for tasks in DMControl, we report the maximum reward score, rescaling to be in the range of [0, 100] by dividing by 10. We report the average metric across tasks for each environment. In addition, it is worth noting that the metrics we report are the **average value across all base seeds and camera viewpoints**. Finally, we also report the results of our post-pre-training STP (ViT-B/16) on each task in Table 8.

In addition, we emphasize that different random seeds primarily affect the rendering of the initial frame in the sampled trajectories, as shown in Figure 7. During evaluation, the seed value we provide serves as the base seed, and the trajectory sampling process is depicted in Algorithm 1. **Therefore, the actual number of trajectories we evaluate is the number of trajectories multiplied by the number of base seeds.** For instance, for MetaWorld, we evaluate $25 \times 3 = 75$ trajectories, with random seeds for rendering being 100-124, 200-224, and 300-324.

Finally, for Franka-Kitchen, we utilize MuJoCo210, while all other simulation environments are based on MuJoCo200. Our policy training and evaluation environments are conducted on Cuda 11.3, NVIDIA TITAN Xp GPUs, and OpenGL 3.1.

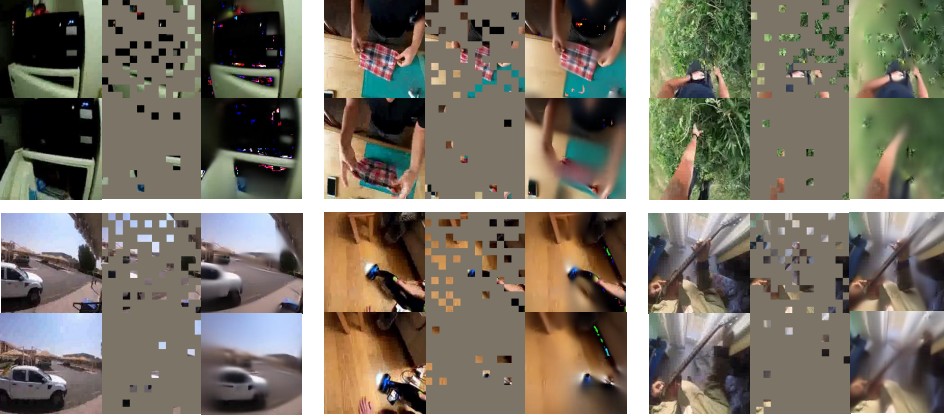

Figure 6: Some examples of our STP prediction result on Ego4D videos. For each six tuple, we show the ground-truth (left), masked frames (middle), STP prediciton results (right), current frames (top), and future frames (bottom). We simply overlay the output with the visible patches to improve visual quality.

[2]https://github.com/facebookresearch/eai-vc/tree/main/cortexbench
[3]https://github.com/facebookresearch/r3m/tree/eval/evaluation

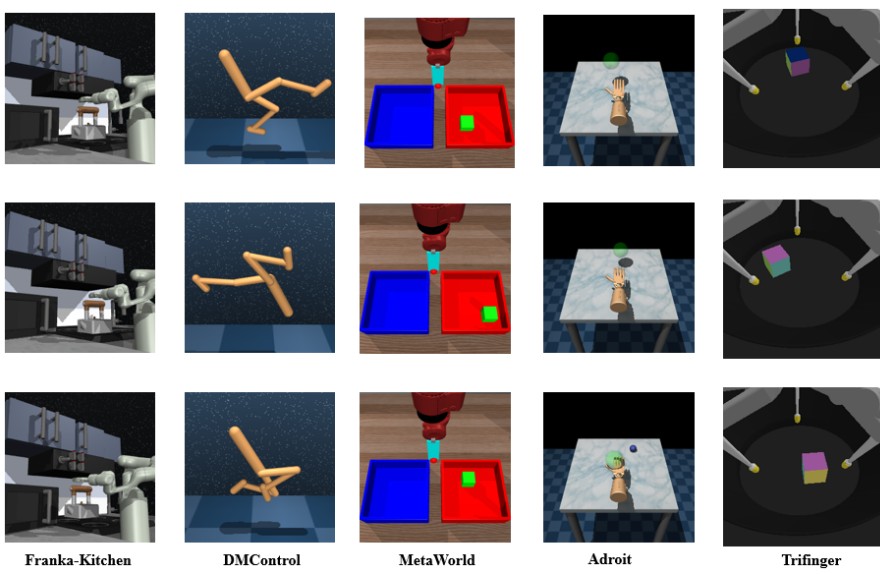

**Franka-Kitchen**    **DMControl**    **MetaWorld**    **Adroit**    **Trifinger**

Figure 7: The visualization of initial frame rendering under different random seeds.

**Algorithm 1** Trajectories Sampling Pseudocode

```
# num_traj: the number of evaluation trajectories
# base_seed: base seed for rollouts

# rollout to sample trajectories
    for ep in range(num_traj):
        seed = base_seed + ep
        env.set_seed(seed)
        o = env.reset()
```

## A.5 Real-World Environments Details

In this section, we outline the details of our real-world setup and evaluation scheme. As depicted in Figure 8, our real-world scenario includes four camera viewpoints: top, left, right, and wrist. It includes two Kinect DK and two RealSense cameras. An example of four views is shown in Figure 9. Specifically, we utilize four different camera views and resize their resolution uniformly to $224 \times 224$. To effectively model the complex and multimodal action distribution in our real-world tasks, we select diffusion policy [16] as our policy model. In accordance with this approach, we concatenate the visual embeddings of all views from two sequential frames. Following the approach in [27], we collect robot data using a VR tele-operation setup. In this way, we collect 100 continuous trajectories for each task. It is worth noting that the quality of these demonstrations leaves room for improvement and contains a lot of noise. During the evaluation phase, we primarily evaluate two contact-rich tasks that have not appeared in Franka-Kitchen [31] benchmark: (1) Picking. It requires the robot arm to pick up the transparent bowel off the table; (2) Pouring. It requires the robot arm to pour at least three-quarter of the ingredients from the transparent bowl into the black pot. For each task, we change the initial pose of the robot arm and objects within a certain range as well as conduct 20 trials. In addition, there are different distractors on the desktop during training and testing, which also evaluates the robustness of the model to distractors. Throughout the process, we use ROS and MoveIt for hardware communication and motion planning.

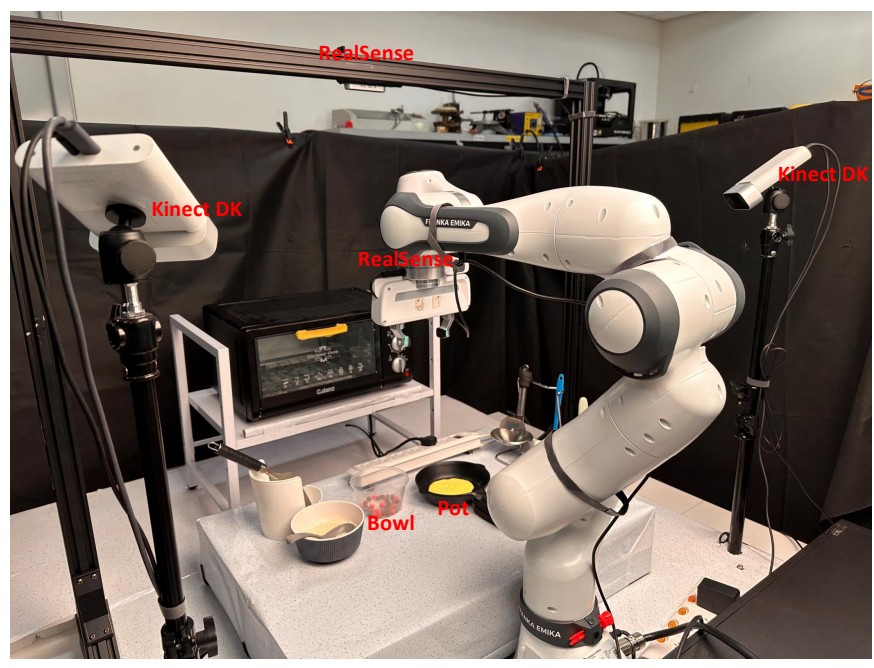

Figure 8: Our real-world scene with four cameras and a Franka Emika robot arm.

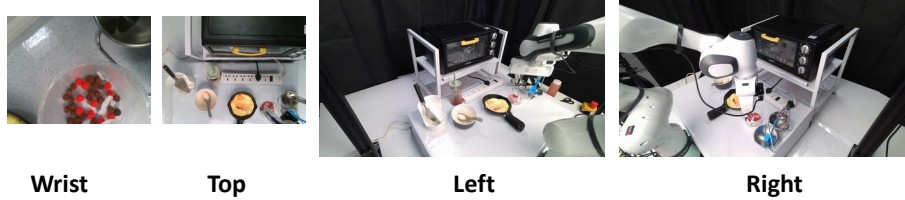

**Wrist**         **Top**                    **Left**                              **Right**

Figure 9: An example of four views.

Table 5: STP post-pre-training hyperparameters on simulation environments.

|  | MetaWorld | Franka-Kitchen | DMControl | Adroit | Trifinger |
|---|---|---|---|---|---|
| total epochs | 50 | 100 | 50 | 50 | 50 |
| warmup epochs | 5 | 5 | 5 | 5 | 5 |
| effective batch size | 1024 | 128 | 2048 | 1024 | 1024 |
| number of demonstrations | 25 | 25 | 100 | 100 | 100 |
| frame interval | 4 | 4 | 4 | 4 | 16 |

Table 6: Policy training hyperparameters on simulation environments.

|  |  | MetaWorld | Franka-Kitchen | DMControl | Adroit | Trifinger |
|---|---|---|---|---|---|---|
| epochs |  | 100 | 480 | 100 | 100 | 100 / 1000 |
| batch size | frozen | 256 | 32 | 256 | 256 | 32 |
|  | fine-tuning | 64 | 32 | 64 | 64 | 16 |
| learning rate | frozen | 0.001 | 0.001 | 0.001 | 0.001 | 0.0001 |
|  | fine-tuning | 0.00005 | 0.0001 | 0.00005 | 0.00005 | 0.0001 |

Table 7: Evaluation schemes on simulation environments.

| Benchmark | Observation Space | History Window Size | Camera ViewPoints | Base Seeds | Number of Trajectories |
|---|---|---|---|---|---|
| Metaworld | RGB + prop. | 3 | top_cap2 | 100, 200, 300 | 25 |
| Franka-Kitchen | RGB + prop. | 1 | left, right | 123, 124, 125 | 50 |
| DMControl | RGB | 3 | 0 | 100, 200, 300 | 25 |
| Adroit | RGB + prop. | 1 | vil_camera | 100, 200, 300 | 25 |
| Trifinger | RGB + prop. | 1 | default | 10 | 25 |

Table 8: The success rate for each task on simulation bechmarks.

| Assembly | Bin-Picking | Button-Press | Drawer-Open | Hammer |
|---|---|---|---|---|
| 94.7 | 97.3 | 94.7 | 100.0 | 100.0 |
| Sliding Door | Turning Light on | Opening Door | Turning Knob | Opening Microwave |
| 96.0 | 72.7 | 39.0 | 31.3 | 29.0 |
| Relocate | Reorient-Pen | Finger-Spin | Cheetah-Run | Reacher-Hard |
| 49.3 | 77.3 | 69.6 | 71.9 | 87.7 |
| Walker-Stand | Walker-Walk | Reach-Cube | Push-Cube | |
| 95.9 | 89.0 | 85.3 | 70.6 | |

