# OpenReview forum: "Spatiotemporal Predictive Pre-training for Robotic Motor Control"
_NeurIPS.cc/2024/Conference — Submitted to NeurIPS 2024_

### Official Review · Reviewer_Dk2S · 2024-06-28

**Soundness:** 2
**Presentation:** 3
**Contribution:** 2
**Rating:** 5
**Confidence:** 3

**Summary:**

This paper studies how to extract useful visual features from out-of-domain and action-free human videos to enhance robotic visualmotor control. Specifically, the authors argure that naively extracting spatial features via MAE is insufficient for robotics control, in contrast, jointly captureing spatial control and temporal movement will be more effective. To do so, the authors propose STP, a new self-supervised learning method, that simutaneously performs MAE on current frame to extract spatial information and predict furture frames to extract temporal motion clues. The overall motivation, idea and method are straightforward and reasonable. The authors evaluate STP on diverse benchmarks including 21 tasks spanning from simulation to real world tasks using imitation learning.

**Strengths:**

1. The paper is well-motivated, highlighting the importance of pre-training visual features for robotic foundation models.

2. The logic in the paper is clear and easy to follow.


3. The proposed method is straightforward and simple to implement.

**Weaknesses:**

1. The high costs associated with evaluating real-world tasks using different random seeds make it challenging to report variances. However, assessing the impact of multiple random seeds in simulated tasks could provide more reliable statistical insights. As shown in Table 1, STP's performance improvement over baselines is marginal (STP 63.7 vs. VC-1 61.8, and STP-L/16(Post PT) 78.4 vs. MAE-L/16(Post PT) 76.7). Given the inherent stochastic nature of imitation learning and reinforcement learning, evaluations across multiple episodes and various random seeds are crucial to validate the proposed methods effectively.

2. Some previous methods also consider the temporal movements when extracting the visual features. For instance, the video-language alignment loss in R3M [1] tries to align language with correct visual transitions, which can extract semantic informations about visual movements. Voltron[2] and DecisionNCE [3] also try to extract the semantic features of the temporal movements between two frames. VIP[3] and LIV[4] use RL to extract visual features, which may also capture long-term movements via bootstrapping. Therefore, the authors could strengthen their paper by highlighting these related works, demonstrating awareness of existing methods, situating their contributions and highlighting the differences between STP and these baselines.

[1] R3M: A Universal Visual Representation for Robot Manipulation. CoRL 2023

[2] Language-Driven Representation Learning for Robotics. RSS 2023.

[3] DecisionNCE: Embodied Multimodal Representations via Implicit Preference Learning. ICML 2024.

[4] VIP: Towards Universal Visual Reward and Representation via Value-Implicit Pre-Training. ICLR 2023.

[5] LIV: Language-Image Representations and Rewards for Robotic Control. ICML 2023

**Questions:**

Please refer to the weakness for details.

**Limitations:**

The authors have properly discussed the limiations in the Appendix.

---

> ### Author Rebuttal · Authors · 2024-08-07
>
> **C1: The high costs associated with evaluating real-world tasks using different random seeds make it challenging to report variances. However, assessing the impact of multiple random seeds in simulated tasks could provide more reliable statistical insights. As shown in Table 1, STP's performance improvement over baselines is marginal (STP 63.7 vs. VC-1 61.8, and STP-L/16(Post PT) 78.4 vs. MAE-L/16(Post PT) 76.7). Given the inherent stochastic nature of imitation learning and reinforcement learning, evaluations across multiple episodes and various random seeds are crucial to validate the proposed methods effectively.**
>
> **R1:** Thanks for your suggestions. To address your concern, we first report results with multiple random seeds, then clarify some confusion points in Tab 1, and finally report extra experiment results with more evaluation.
>
> **(1) We rerun the experiments and report the mean and standard deviation, demonstrating the stability of our experiments.**
>
> For the same BC seed, our policy training is **fully reproducible**, with the only uncertainty being the **slight difference that still exist in MuJoCo rendering** even with the same policy and evaluation seed. Therefore, we **further rerun** our paper's STP-B and MAE-B baseline twice during limited time, obtaining three results and their mean and variance, as shown below. These demonstrate the stability of our experiments. In addition, the rendering in Trifinger is not subject to randomness, hence the results are **fully reproducible**.
>
> |              |  Number of BC seeds × Number of evaluation seeds × Number of runs  |  STP-B  |  MAE-B  |
> |:------------:| :------------:  |:-------:|:-------:|
> | Meta-World   |   25×3×3    |  94.1 93.6 94.1 **93.9±0.2**    |    85.1 84.8 84.3 **84.7±0.3**    |
> | Franka-kitchen | 50×3×3   |   42.5 43.5 43.8 **43.3±0.6**    |    36.7 37.9 37.1 **37.2±0.5**    |
> | DMControl    |   25×3×3    |61.6 60.3 60.7 **60.9±0.5**   |    59.2 60.3 60.3 **59.9±0.5**   |
> | Adroit       |   25×3×3   |47.3 48.7 48.0 **48.0±0.6**   |    43.4 45.3 44.7 **44.5±0.8**    |
> | WA       |   25×3×3   |63.9 63.8 64.1 **63.9±0.3**   |    58.3 59.1 58.7 **58.7±0.3**    |
>
>
> **(2) On confusion of Tab 1.: we use the same pre-training data to conduct a fair comparison between MAE and STP.**
>
> Our STP (EgoClip) utilize less pre-training data than the publicly VC-1 (full Ego4D+MNI). Under the same pre-training data and number of training epochs, our STP-B outperforms MAE-B baseline `(63.7 vs. 59.6)`, which shows a significant improvement of `4.1`.  Note that VC-1 shares the same technique with MAE only with different pre-training data.
>
> **(3) About the performance improvement of ViT-Large (PT).**
>
> We analyze that under the post-pre-training and ViT-Large setting, the representation might be overfitted to the target domain, and the performance improvement tends to saturate. In the future, we may resort to larger pre-training data.
>
> **(4) Our initial evaluation scale was enormous, hence the performance improvement is solid and stable.**
>
> The Number of evaluation episodes = Number of tasks × Number of BC seeds × Number of evaluation seeds × Number of camera views. Therefore, the total number of our evaluation episodes is
> `5×3×25×1 + 5×3×50×2 + 5×3×25×1 + 2×3×25×1 + 2×1×25×1 = 2450`.
>
> |  | Number of BC seeds | Number of evaluation seeds | Number of camera views
> | :-------: | :------: | :---------: | :---------: |
> | Meta-World    | 3   | 25      | 1 |
> | Franka-Kitchen    | 3   | 50       | 2 |
> | DMControl    | 3   | 25       | 1 |
> | Adroit    | 3   | 25     | 1 |
> | Trifinger    | 1   | 25     | 1 |
>
>
> **(5) We add the evaluation results from RLbench, demonstrating the generality of our improvements.**
>
> In addition, we have added a performance comparison between STP and VC-1 on `20` randomly selected tasks in RLBench. We use the advanced RVT [6] as the policy module, training a multi-task policy with only `20` demonstrations per task. Each task is evaluated over `25` episodes, and the results are in the PDF file; our `STP (52.0)` outperforms `VC-1 (43.2)` by `8.8`, further proving the effectiveness of our STP.
>
> [6] Rvt: Robotic view transformer for 3d object manipulation.
>
>
> **C2: The authors could strengthen their paper by highlighting these related works, demonstrating awareness of existing methods, situating their contributions and highlighting the differences between STP and these baselines.**
>
> **R2:** Thank you for your suggestion. We will further strengthen the discussion of these related works and highlight our STP's differences in the revised paper.
>
> Our STP has distinct differences from these works that consider the temporal movements in terms of objectives and techniques. R3M, LIV, and DecisionNCE capture the alignment between language instructions and task progression through contrastive learning for representation learning or reward learning. VIP focuses on self-supervised contrastive learning to pretrain a reward function. Voltron performs language-guided reconstruction for multimodal video representation.
>
> In contrast, our STP is a purely **self-supervised** method through **mask-based generative modeling** for spatiotemporal prediciton, using asymmetric masking and decoder architecture to jointly capture content and motion features. Our method does not require language descriptions. Our generative pre-training paradigm differentiates STP to these contrastive methods from the technique perspective.
>
> In summary, our STP has the following advantages: It is more **general** (learning standard image representation instead of focusing on a specific application), **scalable** (self-supervised learning and plain ViT), **efficient** (high masking ration), and **effective** (asymmetric masking and decoder architecture design ensure joint learning of content and motion features).

---

> > ### Comment · Reviewer_Dk2S · 2024-08-08
> > **More in-depth discussion about Voltron should be provided**
> >
> > Thanks for the efforts and these detailed responses! I acknowledge the evaluation efforts the reviewer made to solidify their claims. However, it seems that this paper shares many similarities with Voltron: `they both use masked reconstruction on current and future frames to extract the temporal as well as the spatial features for downstream robot learning`.
> >
> > In my view, the core differences are three folds:
> > 1. `Settings`: Voltron studies the multi-modal setting, but this paper studies the uni-modal setting.
> > 2. `Methods`: Voltron uses the same mask ratios for both current and future frames, but this paper adopts different ratios.
> > 3. `Methods`:Voltron adopts the same transformer decoder to reconstruct current and future frames, but this paper designs a spatial and temporal decoder to decode them separately.
> >
> > So, for me, it looks like this paper degenerates from more complex multi-modal settings in Voltron to simpler uni-modal setups, with the overall objective (reconstructing both the current and future frames) very similar.  In this sense,  the technical contributions look more like some detailed implementation improvements. Therefore, it would be better for the authors to discuss more on the very similar Voltron.

---

> > > ### Author Response · Authors · 2024-08-08
> > > **Discussion on the difference with Voltron**
> > >
> > > Thanks for your prompt reply to our response and acknowledgement to our evaluation efforts. On the difference of our STP with Voltron, we would like to make the following clarifications and hope this can well address your concern.
> > >
> > > 1. Our STP shares a simpler design than Voltron just as you mentioned from multi-modal setting to uni-modal setting, which makes our STP a more scalable approach than Voltron as uni-modal data is more easily obtained.
> > >
> > > 2. The key difference is that our STP decouples the current frame and future frame for separate modeling, while Voltron employs the MAE-ST (VideoMAE) pre-training to jointly model the whole clip (2 frames). Our decoupled design leads to several important differences in technique design:
> > >
> > > a. The encoder design is different. Our STP encoder processes each frame independently and there is no attention operation between frames. However, the Voltron encoder operates on a clip to learn a video-level representation and there is cross-frame attention operations. Our design aims to encourage our encoder to learn an image-level representation that is temporally-sensitive for prediction. We find this image-level encoder is more friendly for the downstream adaption compared with the video-level representation (video encoder has higher computational cost).
> > >
> > > b. The decoder design is different. Our STP has two decoders: one for spatial prediction and the other for temporal prediction, to treat current frame prediction and future frame prediction separately. For the spatial decoder, we use `joint self-attention` to process the current frame; for the temporal decoder, we add `cross-attention` to capture the interaction between the current frame and future frame (See Fig. 2 in the paper). `This design ensures the predictive property of STP, in sense that the representation of current frame will not see the future and acts as condition for future prediction`. The Voltron only employs a single decoder to reconstruct the whole video directly.
> > >
> > > c. The masking strategy is different. The decoupled design allows STP to assign different masking ratios to current frame and future frame. Specifically, we use ratio of `75%` and `95%` for them, while Voltron uses the same ratios for all frames.
> > >
> > > 3. The ablation studies in in Tab 2(a), Tab 2(b) and Tab 2(c) demonstrate that these different technique designs as mentioned above are `crucial` for achieving excellent performance. Meanwhile, during rebuttal, we add a direct comparison with MAE-ST (The architecture is the same with Voltron without language input). Our STP is better (63.7 vs. 52.6, see response to Reviewer QdYc).
> > >
> > > 4. Finally, our evaluation is more comprehensive than Voltron on the backbone scaling, pre-training data, and robotic motor control downstream tasks. In terms of pre-training data, Voltron only uses `small-scale` Something-Something-v2 dataset for pre-training. At the same time, the model size of Voltron (V-gen) is only `small`. In contrast, STP uses larger scale training data and trains the `ViT-Large`.  Our self-supervised and uni-modal settings ensure the scalability and generality of STP.
> > >
> > > If you have further concerns, please feel free to comment. We would like to answer your question.

---

> ### Comment · Reviewer_Dk2S · 2024-08-08
>
> I agree with the authors that this paper conducts more comprehensive ablations and evaluations to support the effectiveness of each design choices. In this sense, I am very happy to increase my score. Meanwhile, considering the similarity to Voltron (the authors provide more detailed techinique insights about the differences to voltron, but the high-level differences are like what I pointed out). I decide to increase to a 5 (boardline accept).

---

> > ### Author Response · Authors · 2024-08-08
> >
> > Thanks for your comments and the recognition of our responses.

---

### Official Review · Reviewer_QdYc · 2024-07-11

**Soundness:** 3
**Presentation:** 3
**Contribution:** 3
**Rating:** 6
**Confidence:** 4

**Summary:**

The paper presents a new spatio-temporal pretraining algorithm for representation learning for robotics. The authors propose using masked autoencoding for reconstructing the current frame (for spatial reasoning) and a future frame (for temporal reasoning). The authors provide extensive experimentation across simulated and real-world settings and provide ablation studies to justify their design choices.

**Strengths:**

- The paper addresses the important topic of including temporal dynamics in video data for pretraining robot representations.
- The paper does a good job of explaining the method and detailing the various experimental settings.
- The authors provide policy performance using both the pre-trained representations and post-pre-trained representations which helps assess both the quality of representations learned from internet data as well as the advantage of finetuning representations on the task-specific data. Overall, the proposed method has been extensively evaluated over varied settings across a variety of simulated settings.
- The authors provide an insightful ablation study to justify their design choices.

**Weaknesses:**

- It is unclear where the diverse image data for STP trained with Ego+I in Table 1 is obtained from. Some information about this would be helpful.
- The real-world experiments seem limited with only two real-world tasks where the MAE also performs reasonably well.
- The authors must include comparisons with prior works using MAE for spatiotemporal learning [1].

[1] Feichtenhofer, Christoph, Yanghao Li, and Kaiming He. "Masked autoencoders as spatiotemporal learners." Advances in neural information processing systems 35 (2022): 35946-35958.

**Questions:**

It would be great if the authors could address the “Weaknesses” listed above.

**Limitations:**

The limitations have been addressed adequately.

---

> ### Author Rebuttal · Authors · 2024-08-07
>
> **C1: It is unclear where the diverse image data for STP trained with Ego+I in Table 1 is obtained from.**
>
> **R1:** Sorry for the confusion. "STP trained with Ego+I" means that we perform a **hybrid pre-training** using EgoClip and ImageNet data. Specifically, we first initialize ViT with the ImageNet-MAE weight. During the pre-training process, for the image data from ImageNet, we conduct MAE pre-training; for the video data from EgoClip, we conduct STP pre-training. This resultes in a 0.5 performance improvement, indicating that our STP can also benefit from more diverse image data.
>
>
> **C2: The real-world experiments seem limited with only two real-world tasks where the MAE also performs reasonably well.**
>
> **R2:** Our STP has achieved significant advantages in the pouring task (**45% -> 65%**). It can more accurately align with the moving bowl and the pot. In addition, although MAE and STP have a same success rate in picking tasks, STP tends to execute grasping in a better position. Some works such as [2] have demonstrated that real-world environments and simulation settings yield similar conclusions, hence we have not carried out more and costly real-world evaluations. We will release our STP weight in the future, hoping that it could contribute to the community and be applied to more real-world environments and tasks.
>
> [2] What do we learn from a large-scale study of pre-trained visual representations in sim and real environments?
>
> **C3: The authors must include comparisons with prior works using MAE for spatiotemporal learning [1].**
>
> **R3:** Thanks for your suggestion. Our STP differs from MAE-ST as STP executes **asymmetric** masking and decoder architecture design for **decoupled** spatial and temporal prediction on the **2D image model**. On the contrary, MAE-ST jointly performs spatiotemporal reconstruction to pre-train the **3D video model**, processing the temporal dimension and spatial dimension **symmetrically**.
>
> We pre-trained a 4-frame MAE-ST based on the EgoClip dataset, and the results are shown below. We believe that the poorer performance of MAE-ST is due to the gap in temporal interaction between upstream and diverse downstream environments, which lead to a significant risk of cumulative error in the paradigm of imitation learning.
>
> |  | Meta-World | Franka-Kitchen   | DMControl   | Adroit   | Trifinger   | WA |
> | :---: | :---:    | :---:   | :---:   | :---:   | :---:   | :---:   |
> | STP (EgoClip)| 92.0   | 40.9   | 62.1   |  48.0  |   69.3 |  **63.7**  |
> | MAE-ST (EgoClip) |  68.5  | 30.5   | 53.9  |   47.3 |  70.5  |  52.6  |

---

> > ### Comment · Reviewer_QdYc · 2024-08-08
> > **Thank you for the clarifications**
> >
> > I thank the authors for the clarifications and the additional results. After considering the rebuttal, I would like to keep my score.

---

> > > ### Author Response · Authors · 2024-08-08
> > > **Thanks for your positive response**
> > >
> > > Thanks for your quick reply and acknowledgement to our response.

---

### Official Review · Reviewer_UXtm · 2024-07-16

**Soundness:** 3
**Presentation:** 3
**Contribution:** 2
**Rating:** 6
**Confidence:** 4

**Summary:**

This paper proposes STP, a visual representation learning method for robotic motor control. Trained on human videos, STP uses masked auto-encoders for spatial-temporal prediction. The spatial decoder predicts the current frame from its representation with 75% of patches masked. The temporal decoder predicts the future frame using the representations of 75%-masked current frame and the 95%-masked future frame. Experiments on various simulation and real-world tasks show the effectiveness of STP compared with baselines.

**Strengths:**

1. The proposed method is simple yet effective, utilizing a masked spatial-temporal prediction objective to learn visual representations for robotics.
2. The paper presents extensive experimental results in both simulation and real-world settings, comparing with proper visual representation baselines.

**Weaknesses:**

1. Many works have considered temporal information for robot visual representation learning. This paper should mention these and highlight the differences. For example, R3M [1] uses temporal contrastive learning, while VIP [2] and V-PTR [3] use temporal difference.
2. Though STP outperforms the baselines in many benchmarks, the performance gap is not significant (Table 1). The slight performance difference may be due to hyperparameter selection and randomness, as the paper did not provide error bars over multiple seeds.

[1] R3m: A universal visual representation for robot manipulation, 2023
[2] Vip: Towards universal visual reward and representation via value-implicit pre-training, 2022
[3] Robotic Offline RL from Internet Videos via Value-Function Pre-Training, 2023

**Questions:**

1. I think VIP and V-PTR should be included as baselines.
2. What is the evaluation protocol for downstream tasks? Does all evaluation use an expert dataset and perform imitation learning to learn a policy? How did you collect the dataset for real-world experiments?

**Limitations:**

The authors have discussed the limitations. These cannot be addressed within the scope of this paper.

---

> ### Author Rebuttal · Authors · 2024-08-07
>
> **C1: This paper should highlight its differences with R3M, VIP, and V-PTR.**
>
> **R1:** Thank you for your suggestion. We will highlight these differences in the revised paper. Our STP has distinct differences from these works in objectives and techniques.
>
> VIP and V-PTR respectively pre-train the value function through contrastive learning and TD learning, focusing on **visual reward functions and value function in RL**. R3M pre-trains image representation using **time-contrastive learning** and **video-language alignment**.
>
> In contrast, our STP performs **self-supervised**, masking-based **generative modeling**, which is more **efficient** (high masking ratio), and **scalable** (self-supervised learning and simple backbone of ViT allows for large-scale pre-training).  Our generative pre-training paradigm differentiates us to these contrastive methods. The superior performance of STP demonstrates the advantage of generative pre-training against them.
>
> **C2: The performance gap is not significant. The slight performance difference may be due to hyperparameter selection and randomness, as the paper did not provide error bars over multiple seeds.**
>
> **R2:** **First, we clarify some confusions on the result in Tab 1.**
>
> **(1) In a fair comparison, our improvement is significant.**
>
> Our STP (EgoClip) utilize less pre-training data than the VC-1 (full Ego4D+MNI). Under the same pre-training data and the number of training epochs, our STP-B outperforms MAE-B baseline `(63.7 vs. 59.6)`, which shows a significant improvement of `4.1`. Note that VC-1 shares the same technique with MAE only with different pre-training data.
>
> **(2) Our initial evaluation scale was enormous, hence the performance improvement is stable.**
>
> The Number of evaluation episodes = Number of tasks × Number of BC seeds × Number of evaluation seeds × Number of camera views. The total number of our evaluation episodes is
> `5×3×25×1 + 5×3×50×2 + 5×3×25×1 + 2×3×25×1 + 2×1×25×1 = 2450`.
>
> |  | Number of BC seeds | Number of evaluation seeds | Number of camera views
> | :-------: | :------: | :---------: | :---------: |
> | Meta-World    | 3   | 25      | 1 |
> | Franka-Kitchen    | 3   | 50       | 2 |
> | DMControl    | 3   | 25       | 1 |
> | Adroit    | 3   | 25     | 1 |
> | Trifinger    | 1   | 25     | 1 |
>
> **Second, we rerun the experiments with multiple seeds and report the mean and standard deviation.**
>
> During pre-training and BC, we did not deliberately select hyperparameters. The pre-training hyperparameters of STP follow MAE, and all representations adhere to the same setting during BC. For the same BC seed, our policy training is **fully reproducible**, with the only uncertainty being the **slight difference that still exist in MuJoCo rendering** even with the same policy and evaluation seed. Therefore, we **further rerun** our paper's STP-B and MAE-B baseline twice during rebuttal, obtaining three results and their mean and variance, as shown below. These demonstrate the stability of our STP. Additionally, the rendering in Trifinger is not subject to randomness, hence the results are **fully reproducible**.
>
> | |  Number of BC seeds × Number of evaluation seeds × Number of runs  |  STP-B  |  MAE-B  |
> |:-:| :--:  |:-:|:-:|
> | Meta-World  |   25×3×3    |  94.1 93.6 94.1 **93.9±0.2**  |    85.1 84.8 84.3 **84.7±0.3**    |
> | Franka-kitchen | 50×3×3   |   42.5 43.5 43.8 **43.3±0.6**  |    36.7 37.9 37.1 **37.2±0.5**    |
> | DMControl|   25×3×3    |61.6 60.3 60.7 **60.9±0.5**  |    59.2 60.3 60.3 **59.9±0.5**   |
> | Adroit|   25×3×3   |47.3 48.7 48.0 **48.0±0.6**   |    43.4 45.3 44.7 **44.5±0.8**    |
> | WA |   25×3×3   |63.9 63.8 64.1 **63.9±0.3**   |    58.3 59.1 58.7 **58.7±0.3**    |
>
> **Finally, we add extra evaluation results from RLbench, demonstrating the generality of our improvements.**
>
> In addition, we added a performance comparison between STP-B and VC-1 base on `20` randomly selected tasks in RLBench. We use the advanced RVT [5] as the policy, training a multi-task policy with only `20` demonstrations per task. Each task is evaluated over `25` episodes, and the results are in the PDF file; our `STP (52.0)` outperforms `VC-1 (43.2)` by `8.8`, further proving the effectiveness of our STP.
>
> **C3:  VIP and V-PTR should be included as baselines.**
>
> **R3:** Thanks for your suggestions. Since V-PTR has not released weight, we report the results for VIP and LIV [4] as baseline comparison. It is worth noting that both VIP and LIV utilize ResNet50 as their backbone, making a direct comparison with ViT-B is **unfair**. Additionally, the difference in feature dimensions between ResNet50 and ViT-B (2048 vs. 768) results in a discrepancy in the number of trainable MLPs policy parameters. Therefore, to enable a **a more fair comparison**, we construct the policy with an equivalent number of parameters for STP-B, which we refer to as STP†, and the results demonstrate the superior performance of our STP.
>
> |  | Meta-World | Franka-Kitchen   | DMControl   | Adroit   | Trifinger   | WA |
> | :-: | :-:    | :-:   | :-:   | :-:   | :-:   | :-:   |
> | VIP | 86.4   | 38.1   |  70.5  |  55.3  |  68.9  |  64.4  |
> | LIV| 81.3   | 37.3   | 54.0   |  52.0  |   68.3 |   58.1 |
> | STP† |  93.6  | 44.0   |  69.7  |   51.3 |   67.9 |  **67.1**  |
>
> **C4: What is the evaluation protocol for downstream tasks? Does all evaluation use an expert dataset and perform imitation learning to learn a policy?**
>
> **R4:** We follow a single-task setup due to without task condiiton. Yes, for each task, we learn a single policy based on the representation. In extra RLbench experiments, we use the multi-task setup.
>
> **C5: How did you collect the dataset for real-world experiments?**
>
> **R5:** As shown in A.5, following the approach in [6], we collect robot data using a VR tele-operation setup.
>
> [4] LIV: Language-Image Representations and Rewards for Robotic Control.
> [5] Rvt: Robotic view transformer for 3d object manipulation
> [6] Openvr: Teleoperation for manipulation.

---

> > ### Comment · Reviewer_UXtm · 2024-08-12
> >
> > Thank you for your reply. I appreciate your efforts in running experimental results for baselines and for multiple seeds. While STP underperforms compared to VIP in some benchmarks, it excels in others. I have decided to raise my score to 6.

---

> > > ### Author Response · Authors · 2024-08-13
> > > **Thanks for your comments**
> > >
> > > Thanks for your comments and the recognition of our responses.

---

### Official Review · Reviewer_hmK3 · 2024-07-18

**Soundness:** 3
**Presentation:** 3
**Contribution:** 2
**Rating:** 5
**Confidence:** 4

**Summary:**

In this paper, we present a self-supervised pre-trained visual representation in robotic motor control, with spatiotemporal prediction with dual decoders, utilizing large-scale video data. The spatial prediction follows a standard MAE pipeline, and the temporal prediction tries to predict the future based on the current frame. The trained encoder is applied to downstream tasks and real-world robot task for better sample efficiency.

**Strengths:**

1. This paper adopts actionless human video data for representation learning, which can be easily obtained. The learned representation can be adapted to downstream robotics tasks.

2. The experiments contain several real-world tasks, which could be more valuable for applying a pre-trained visual encoder to real-world domains that lack data.

**Weaknesses:**

1. The major concern is the novelty of the previous methods, considering several related papers that leverage human data and visuals pertaining to downstream tasks have been proposed [1-3].

2. The experiment only contains imitation learning experiments in downstream tasks, while the reinforcement learning framework with sub-optimal data is not considered.

[1] Learning Manipulation by Predicting Interaction. RSS 2024

[2] Large-Scale Actionless Video Pre-Training via Discrete Diffusion for Efficient Policy Learning. https://arxiv.org/html/2402.14407

[3] Unleashing Large-Scale Video Generative Pre-training for Visual Robot Manipulation. https://arxiv.org/abs/2312.13139

**Questions:**

N/A

---

> ### Author Rebuttal · Authors · 2024-08-07
>
> **C1: The major concern is the novelty of the previous methods, considering several related papers that leverage human data and visuals pretraining to downstream tasks have been proposed [1-3].**
>
> **R1:** Thanks for your comments. Our STP exhibits some essential  differences  or advantages with these works as follows.
>
> As for [2] and [3], our STP has different motivation and techniques. Both [2] and [3] perform a **history-aware policy pre-training** through **language-driven video prediction**, where [2] uses VQ-VAE and video diffusion techniques, while [3] employs auto-regressive GPT-style techniques based on frozen ViT representation. In contrast, our STP performs **image representation pre-training** through joint spatiotemporal prediction in a **self-supervised manner**, using **asymmetric masking and decoder architecture design**. Representation pre-training is orthogonal to these two methods.
>
> As for [1], it pre-trains visual representations by predicting the transition frame and detecting the interaction object. It requires language and bounding box annotations, using only 93K video clips for pre-training. Instead, our STP is a purely self-supervised method without language or object annotations. Our STP is a much simpler design without the multi-frame causality modeling, multimodal token aggregator, and multiheaded attention pooling from [1], and only uses only ViT backbone. The following fair comparison (both use ViT-B) indicate that our STP achieves stronger performance, thanks to the scalability (masking and self-supervised learning enable lager scale data) of our proxy task, the asymmetric masking strategy, and specific decoder architecture design for spatiotemporal prediction.
>
> |  | Meta-World | Franka-Kitchen   | DMControl   | Adroit   | Trifinger   | WA |
> | :---: | :---:    | :---:   | :---:   | :---:   | :---:   | :---:   |
> | STP-B | 94.1   | 42.5   |   61.6 |  47.3  |  66.7  | **64.2**   |
> | MPI-B | 82.1   | 38.4   |  55.7  | 49.3   |   67.7 |   **58.7** |
>
> In summary, our STP makes **orthogonal contributions** to [2] and [3]. Compared to [1], it has the following advantages: It is more **general** (learning standard ViT representation), **scalable** (self-supervised learning), **efficient** (high masking ration), and **effective** (asymmetric masking and decoder architecture design for spatiotemporal prediction). Finally, thank you for your reminder, and we will strengthen these distinctions in the revised paper.
>
> **C2: The experiment only contains imitation learning experiments in downstream tasks, while the reinforcement learning framework with sub-optimal data is not considered.**
>
> **R2:** Thanks for your suggestion. In rebuttal, we select the `Panda-Door` and `Panda-TwoArmPegInHole` tasks from the Robosuite [4] simulation environment for reinforcement learning evaluation. We employ DrQ-v2 as our RL algorithm and compare the results of VC-1 (ViT-B) and our STP (ViT-B) as frozen visual representations. Due to the large fluctuations in success rate, we report the maximum reward value under 200,000 steps, and the results preliminarily verify the effectiveness of our STP within the RL framework.
>
> |  | Panda-Door | Panda-TwoArmPegInHole
> | :---: | :---:    | :---:   |
> | STP | **95.6**   | **130.7**   |
> | VC-1 | 88.8   | 123.1   |
>
> [4] A modular simulation framework and benchmark for robot learning.

---

> > ### Comment · Reviewer_hmK3 · 2024-08-13
> >
> > Thanks for the response. The sub-optimal setting with RL requires further investigation. I keep the original evaluation.

---

> > > ### Author Response · Authors · 2024-08-13
> > >
> > > Thank you for your comments and acknowledgement of our responses. We primarily follow the setting of a series of previous works [1,5,6,7,8], employing a **computation and data-efficient** paradigm of **few-shot behavior cloning by learning from demonstrations (Lfd)** to verify the effectiveness of the visual representation. After carefully considering your suggestions, we have reported some preliminary reinforcement learning results in the review, and we will further explore a more comprehensive evaluation in the future. We hope that the insights gained from our experiments will contribute to further improvements in reinforcement learning for robotics motor control in future work.
> > >
> > > [5] The (Un)Surprising Effectiveness of Pre-Trained Vision Models for Control.
> > >
> > > [6] Real-World Robot Learning with Masked Visual Pre-training.
> > >
> > > [7] Language-driven representation learning for robotics.
> > >
> > > [8] An unbiased look at datasets for visuo-motor pre-training.

---

### Author Rebuttal · Authors · 2024-08-07

We thank all reviewers' efforts in reviewing our paper and giving insightful comments and valuable suggestions. The reviewers' main concerns are concentrated on two primary issues, which we have addressed individually.

**1. There should be a more in-depth discussion on the difference of STP with other works that perform robotics pre-training using video data (hmK3, UXtm, and Dk2S).**

We further strengthen the differences analysis of our STP in terms of contributions, techniques,  and results. We emphasize from the following aspects:

- **Contributions & Differences:** Our goal is to present a genreal and scalable representation pre-training method for robotic motor control without any supervision siginal. However, several existing methods all need language annotations. In addition, our design is simple and efficient with a plain ViT. It is not a video or multimodal representation (Voltron) and does not have any intricate structures (MPI). Furthermore, we will not focus on certain specific applications (VIP, V-PTR), nor will we design policy pre-training based on a specific policy architecture (GR-1, VPDD).

- **Technique**: Different from contrastive pre-training (VIP, R3M), STP adopts masked generative pre -training. Our STP uses asymmetric masking and decoder architecture design to conduct decoupled spatiotemporal prediction, jointly capturing content and motion features.

- **Experiment and Comparison**: We carry out the largest-scale BC evaluation of PVRs for robotic motor control to demonstrate the effectiveness of STP and yield some insightful observations. In a fair comparison, the average performance of our STP is superior to existing representations.


**2. There may be a perceived stability concern regarding the performance improvement of STP over the MAE baseline (UXtm and Dk2S).**

 We detailed the scale of our extensive evaluation (a total of `2450` eposides). and also provide further rerun evaluations, reporting the **mean and standard deviation**, demonstrating that our improvement is **solid and stable**. In addition, we further evaluate the multi-task setting on extra `20` tasks in the RLBench simulation environment, with detailed results in the attached PDF file. Our STP shows an `8.8` improvement in success rate compared to VC-1 (`from 43.2 to 52.0`), which further demonstrates the effectiveness of STP.

---

### Decision · Program_Chairs · 2024-09-25

**Decision:**

Reject

**Comment:**

The paper presents a spatio-temporal pretraining pre-training method, STP, for robotic motor control. While the authors have addressed initial concerns with additional experiments, there are still concerns about the approach's novelty and the marginal performance improvements over existing techniques. A significant issue is the need for more extensive and varied real-world robotic experiments. Considering the feedback, the paper's limited distinction from existing methods, modest performance improvements, and the need for more comprehensive real-world robotic testing, the manuscript does not meet the acceptance criteria. The decision is to reject the paper.